# Extracellular vesicle release in an experimental ventilator-induced lung injury porcine model

Niklas Larsson[1]*, Jonas Claesson[1], Stefan Lehtipalo[1], Annelie Behndig[2],
Fariborz Mobarrez[3], Michael Haney[1]

1 Department of Diagnostics and Intervention, Anesthesiology and Intensive Care Medicine, Umeå University, Umeå, Sweden, 2 Department of Public Health and Clinical Medicine, Umeå University, Umeå, Sweden, 3 Department of Medical Sciences, Clinical Chemistry, Uppsala University, Uppsala, Sweden

* niklas.larsson@umu.se

## Abstract

Harmful effects of mechanical ventilation with large tidal volumes, volutrauma, may contribute much to diffuse acute lung injury. Extracellular vesicles have been noted in the context of vital organ injury. We hypothesized that extracellular vesicles from acutely injured lung can be found in both lung and blood. In a two-hit experimental porcine model, we tested if extracellular vesicles could be detected in bronchoalveolar lavage fluid and in plasma over a six-hour period of large tidal volume ventilation after surfactant depletion. After 2 hours of volutrauma, bronchoalveolar lavage fluid showed increased levels of extracellular vesicles containing nucleic acids (stained by SYTO 13) and those positive for both SYTO 13 and HMGB1. No such increase was detected in plasma at any timepoint during the six-hour experiments. This shows that nucleic acid-containing extracellular vesicles appear to be involved in progression of lung injury, possibly indicating cellular damage, but their potential to serve as diagnostic biomarkers of acute lung injury progression, based on plasma sampling, and in the very early phase, is not confirmed by these findings.

## Introduction

Positive pressure ventilatory support is an indispensable tool in the treatment of severe acute lung injury and ARDS and also in other forms of advanced respiratory failure in the intensive care unit (ICU). Mechanical ventilation, through physical stretch injury to lung parenchyma, can cause serious injury to the lungs, especially when large tidal volumes are applied [1,2]. Diseased lungs appear to be more susceptible to ventilator-induced lung injury (VILI) [3]. Despite overwhelming evidence for benefits of low tidal volume ventilation, many patients with or without lung disease still receive mechanical ventilation with tidal volumes large enough that stretch-related injury is occurring [4]. One of the clinical challenges is that it is difficult to measure, at the bedside, if VILI is occurring. The confirmation of VILI is generally made based on later sequelae.

VILI results from cyclic overdistension of aerated alveoli, and repetitive closure and reopening of small airways [5,6]. This can result in a profound inflammatory process with influx of neutrophils, local and systemic release of inflammatory mediators, alveolar flooding,

**Data availability statement:** All relevant data are within the paper and its Supporting Information files.

**Funding:** This work was supported by the ALF-LUA cooperative Region Västerbotten-Umeå University intramural research funding system. The funders had no role in study design, data collection and analysis, decision to publish, or preparation of the manuscript.

**Competing interests:** Niklas Larsson has received speaker fees for lecturing in clinical symposia sponsored by Dräger Medical, unrelated to this specific topic.

fibrin deposition, and activation of coagulation. The clinical sequelae can be severely impaired respiratory mechanics and gas exchange, as well as systemic inflammation that may be complicated by multiple organ dysfunction syndrome (MODS) [7]. In a clinical setting, with careful attention to positive pressure ventilatory support settings, VILI can be slow developing and difficult to measure at the bedside. The inflammatory processes in the development of VILI, with subsequent tissue damage, seem to be dependent on a number of different signalling pathways, though currently these signalling pathways are not completely understood [8].

Extracellular vesicles (EVs) are cell-derived lipid bilayer membrane-bound particles up to 1,000 nm in size. They are released from a variety of cell types and may contain proteins, various signalling molecules, and nucleic acids such as RNA and DNA from their parent cell [9]. EVs carry antigens on their membranes and cargo that can indicate cell origin and potentially mediate cell signalling [10]. While EVs are released under normal physiological conditions, they are also released in response to stress or injury and can serve as markers of cellular responses and possibly adaptation to damage. High Mobility Group Box 1 (HMGB1) is a nuclear protein functioning as a damage-associated molecular pattern (DAMP) molecule, crucial for inflammation and immune responses [11]. HMGB1 could be associated with EVs, indicating cellular stress or damage. SYTO 13 is a fluorescent nucleic acid stain (binds to DNA and/or RNA) that could indicate damage and stress-related cellular processes [12]. The co-localization of SYTO 13 and HMGB1 in EVs can reflect cellular responses to stress and injury, potentially revealing mechanisms of cell communication and the role of EVs as biomarkers for inflammation-related diseases. Additionally, CD14 positive EVs have been noted in clinical ARDS development [13].

In this study, we focus on EVs released into lung alveoli as well as into blood in relation to an experimental acute lung injury. It is unclear to what extent injurious mechanical ventilation can rapidly induce EV production. To investigate this, we serially sampled both bronchoalveolar lavage fluid (BALF) and blood in a porcine model of VILI. The primary hypothesis was that high tidal volume positive pressure mechanical ventilation to induce acute lung injury will lead to EV release in the first six hours detectable in both bronchoalveolar lavage fluid and blood. The secondary hypothesis is that there is a relation between EV level changes over time in BALF and blood.

## Materials and methods

### Animals

Ethical approval was granted by Umeå Animal Experimental Ethics Committee (A43-12).

The study was conducted with guidance from the EU Directive 2010/63/EU for animal experiments. All subjects were juvenile female Yorkshire/Swedish landrace pigs, bred for the purpose from Forslunda Agricultural School, Umeå, Sweden.

### Preparation

Details of the preparation have already been presented in an earlier report [14]. Premedication included intramuscular ketamine 10 mg/kg (Ketalar® 10 mg/ml, Pfizer AB, Sweden), xylazine 2.2 mg/kg (Rompun® vet 20 mg/ml, Bayer Animal Health, Denmark), and atropine sulphate 0.05 mg/kg (Atropin® Mylan 0.5 mg/ml, Mylan AB, Sweden). Anesthesia was induced with intravenous (IV) pentobarbital 10 mg/kg (Pentobarbitalnatrium, Apoteksbolaget, Stockholm, Sweden) and maintained with fentanyl 20 µg/kg/h (Fentanyl, Braun, Melsungen, Germany), midazolam 0.3 mg/kg/h (Dormicum, Roche, Basel, Switzerland), and pentobarbital 5 mg/kg/h. With orotracheal intubation, mechanical ventilation (Evita 4, Dräger, Kiel, Germany) was performed using 8 mg/kg tidal volume, inspiratory:expiratory ratio (I:E) 1:2, positive end expiratory pressure (PEEP) 5 cmH$_2$O, fractional inspired oxygen 0.4 and respiratory rate set

for normocapnia. There was active external warming with a gel pad set to maintain core temperature at 38°C. Ringer's acetate (Ringer-Acetat® Baxter Viaflo, Baxter Medical AB, Sweden) was infused at 4-5 ml/kg/h. Where hypovolemia was observed, 250 ml of hydroxyethyl starch solution (Voluven®, Fresenius-Kabi, Uppsala, Sweden) was infused. A central venous catheter and an arterial catheter were placed after a cut down in the neck.

## Surfactant depletion

The experimental preparation is described in detail in an earlier publication [14]. Surfactant depletion was performed in all animals (in both experimental groups) using saline lavage as part of a "two-hit" lung-injury model. Preoxygenation was performed with 100% oxygen before lung lavage during brief apnea with 30 ml/kg of 38°C 0.9% saline solution instilled through the tracheal tube before being allowed to drain out passively. The lavage was performed four times, with a short period between each for the arterial oxygen saturation to recover to at least 96%. This was completed before randomized allocation to either large tidal volume ventilation and hyperinflation injury or to low tidal volume ventilation, aiming to avoid hyperinflation injury.

## Ventilatory intervention

After lung lavage, animals were randomly (by blinded lot) allocated to either low (8 ml/kg) tidal volume ventilation (LTV) with PEEP 8 cm $H_2O$ or high (20 ml/kg) tidal volume ventilation (HTV) with zero PEEP. Respiratory rate in the low tidal volume group was adjusted to achieve normocapnia, and inspired oxygen was adjusted achieve arterial oxygen saturation above 96%. The high tidal volume ventilation group had the respiratory rate set to 20 breaths per minute and extra dead space in the breathing circuit was added as needed to achieve normocapnia. All animals in the high tidal volume group were ventilated with 100% inspired oxygen. Animals were ventilated by protocol for six hours, then after the last planned data collection point were euthanized using potassium chloride.

Samples of bronchoalveolar lavage fluid and blood were collected at predetermined points in the protocol (Fig 1). Lung biopsies were collected postmortem and analyzed histopathologically for degree of lung injury, as reported for these subjects in an earlier publication [14,15], in order to verify degree of acute lung injury in the groups.

## Sampling procedures

**Bronchoalveolar lavage.** A flexible fiberoptic bronchoscope, through the tracheal tube and wedged in a bronchus, was used for bronchoalveolar lavage fluid (BALF) collection,

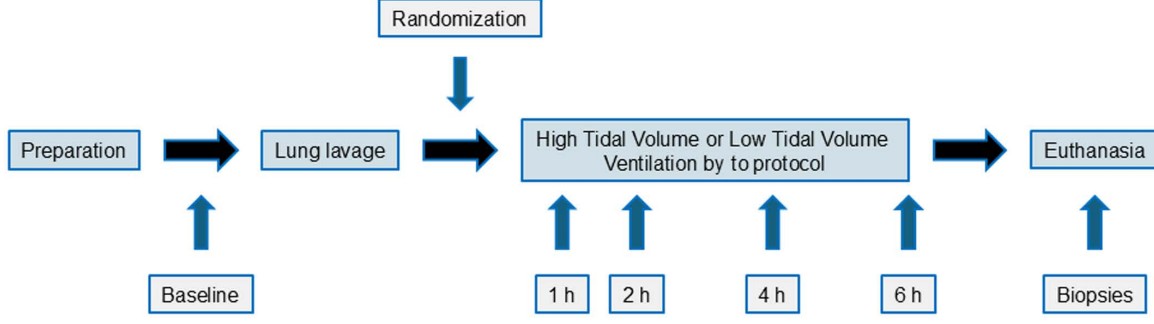

**Fig 1. Study flow chart.** Baseline samples, both bronchoalveolar lavage and arterial blood, were collected in all animals. Lung lavage was performed, followed by randomization to either low tidal volume ventilation or high tidal volume ventilation for 6 hours. Sampling was performed at 1 h, 2 h, 4 h and 6 h of ventilation after lung lavage in both groups.

which meant recovery of fluid exposed to alveoli and distal airways. Three 50 ml aliquots of 0.9% saline solution were administered with retrieval by gentle suctioning. These were combined to make a single sample in an iced glass container. For each animal, the serial sampling was performed in different lung regions. The volume of the recovery was noted after filtration through a 100 μm nylon filter. Each sample was centrifuged for 15 minutes, 4°C at 400 relative centrifugal force (RCF). The supernatant was separated from the cell pellet and stored at −70°C in aliquots.

**Blood samples.** Blood samples were collected in citrate vacutainer tubes and centrifuged at 2000 x g for 20 minutes at room temperature (RT) before the plasma was stored at −70°C for later analysis.

## Laboratory procedure

An EV-enriched pellet was obtained from all samples by high-speed centrifugation [16]. Briefly, 500 μL of PPP was thawed and centrifuged at RT (2000 RCF for 20 minutes). Then, 450 μL of the supernatant was centrifuged at 20,800 RCF for 45 minutes to obtain an EV-enriched pellet. This pellet was diluted with PBS buffer (pH 7.4) and then re-centrifuged at 20,800 RCF for 45 minutes at RT. The final EV-enriched pellet was resuspended in a final volume of 50 μL of plasma. The BALF samples were handled with the exact same protocol as the plasma samples.

20 μL of the EV-enriched pellet, obtained from either plasma or BALF, was incubated for 20 minutes in the dark with 5 μL of lactadherin-FITC (Haematologic Technologies, Vermont, USA), 5 μL of CD61-PC7 (Beckman Coulter, Brea, CA, USA), 5 μL of CD62P/E-APC (Abcam, Cambridge, UK), 5 μL of CD45-APC (Beckman Coulter, Brea, CA, USA), 5 μL of CD14-PE (Beckman Coulter, Brea, CA, USA), SYTO 13 (Invitrogen, Waltham, MA, USA), and HMGB1-PE (Invitrogen, Waltham, MA, USA), or combinations of these markers. EVs were measured by a Beckman Gallios flow cytometer (Beckman Coulter, Brea, CA, USA). The EV gate was determined using Megamix beads (BioCytex, Marseille, France) with diameters of 0.5 μm, 0.9 μm, and 3.0 μm. EVs were defined as particles less than 1.0 μm in size and positive for lactadherin. Conjugated isotype-matched immunoglobulins (IgG1-FITC, IgG1-PE, IgG1-APC, and IgG1-PC7; all from Abcam, Cambridge, UK) without reactivity against porcine antigens were used as negative controls, along with florescence minus one (FMO) controls. Results are presented as EVs/l in the 20 μl EV-pellet. The intra- and inter-assay coefficients of variation for EV measurement were less than 10%.

## Analysis and statistics

A Shapiro-Wilk test was used to assess an approximately normal distribution for repeated measurements of cells and EVs (IBM SPSS Statistics, Version 22. Armonk, NY: IBM Corp.). Characteristics of study groups were compared using Mann Whitney U test for continuous variables. Base line means of study variables were compared between treatment groups using two tailed independent groups t-test (IBM SPSS). For serial measurements and treatment group comparisons, a two-way ANOVA for repeated measures was used to identify treatment effects over time and by group. This was done after performing a one-way ANOVA for each variable and treatment group to detect changes over the serial measurements. ANOVAs of parameters failing to pass the assumption of normality were repeated after transforming the variable according to optimal transformation defined by Box-Cox analysis using the MASS package in R 4.3.2 with the command line *output variable<-boxcox(lm(metabolite~treatment group*time point,data=data set))*. Transformation did not significantly alter the results of any analysis and the untransformed results were thus kept in the report.

## Results

### Characteristics of study groups

15 animal subjects were included in the preparation and all except three completed all the serial sampling points (one animal in the HTV group did not complete the two final sampling points and two animals, also in the HTV group, did not complete the final sampling point). Details for the two experimental treatments are shown in Tables 1 and 2; the HTV group received higher $FiO_2$, larger tidal volumes, higher inspiratory pressures, as well as lower respiratory rate compared to the LTV group. Oxygenation was impaired in the HTV group. Arterial carbon dioxide levels were not different between the groups, though pH was lower in the HTV group. Average total fluid administration per hour was higher in the HTV group.

For BALF and cell analysis at baseline, group comparisons showed no differences (independent samples t-test) for any variable. There was good recovery of instilled lavage fluid with alveolar contents in both treatment groups and at the different sampling points over six hours (Table 3). Neutrophil count in the lavage fluid increased over time for the HTV group (Fig 2). Macrophage counts decreased in both groups.

In BALF, EVs staining positive for genetic material content (SYTO 13 positive) increased over time in the HTV group (p = 0.004) but not in the LTV group, and the HTV group over time was higher than the LTV group (p = 0.02) (Table 4 and Fig 3). EVs positive for both

**Table 1. Treatment variables.**

| Variables | LTV n = 6 | HTV n = 9 | p |
|---|---|---|---|
| Weight (kg) | 35.5 (28–50) | 32.5 (30 – 37) | 0.78 |
| Tidal volume (ml) | 280 (228–400) | 680 (600– 747) | <0.001 |
| Respiratory rate (breaths/min) | 30 (26 – 38) | 20 (20 – 20) | 0.003 |
| Total intravenous crystalloid (ml) | 2000 (975– 3000) | 2500 (2000– 3000) | 0.53 |
| Total intravenous colloid (ml) | 0 (0– 500) | 500 (250– 500) | 0.145 |
| Average fluid administration (ml/kg/h) | 7.7 (4.4– 8.2) | 10.0 (9– 12.0) | 0.026 |
| Average temperature (°C) | 39.4 (38.3– 39.7) | 38.4 (38.2 –39.1) | 0.22 |

Experimental variables presented as median (interquartile range), and comparisons made using Mann Whitney U test. Abbreviations: LTV – low tidal volume, HTV - high tidal volume. This summary has been previously published [14].

**Table 2. Respiratory variables.**

| Variables | LTV n = 6 | HTV n = 9 | p |
|---|---|---|---|
| Last mean airway pressure (cmH$_2$O) | 11 (10 – 11) | 16 (14–18) | <0.001 |
| Last peak inspiratory pressure (cmH$_2$O) | 22 (20 – 23) | 45(41 - 47) | <0.001 |
| Last compliance (ml/cmH$_2$O) | 14.3 (7.0 –21.7) | 14.9 (9.2– 20.6) | 0.60 |
| FiO$_2$ | 0.35 (0.29– 0.40) | 1.0 (1.0– 1.0) | <0.001 |
| Worst P/F ratio (kPa) | 46 (34– 52) | 9.7 (7– 19) | 0.008 |
| Last PCO$_2$ (kPa) | 4.6 (3.8– 5.0) | 7.0 (5.1– 7.3) | 0.018 |
| Lung injury score (histopathology) | 0.125 (0 – 0.44) | 2.75 (2.5 – 2.75) | 0.044 |

Respiratory outcome variables presented as median (interquartile range), and comparisons made using Mann Whitney U test. Abbreviations: P/F- ratio partial pressure of oxygen in arterial blood (kPa) and simultaneous FiO2, PCO2-partial pressure arterial carbon dioxide, LTV – low tidal volume, HTV - high tidal volume. This summary has been previously published [14].

**Table 3. Bronchoalveolar lavage recovery rate and cells.**

| | | Baseline | 1 hour | 2 hours | 4 hours | 6 hours | p† | p# |
|---|---|---|---|---|---|---|---|---|
| **Recovery (%)** | LTV | 70 (20) | 75 (17) | 74 (17) | 73 (16) | 71 (16) | 0.98 | |
| | HTV | 77 (18) | 69 (8.0) | 65 (14) | 66 (12) | 61 (11) | 0.18 | 0.58 |
| **Total cells/ml x10⁴** | LTV | 41 (16) | 13 (11) | 21 (16) | 19 (11) | 16 (17) | 0.02 | |
| | HTV | 42 (23) | 9.1 (4.5) | 23 (9.7) | 25 (14) | 30 (11) | 0.005 | 0.59 |
| **Macrophages/ml x10⁴** | LTV | 36 (15) | 10 (9.2) | 17 (14) | 18 (10) | 13 (16) | 0.02 | |
| | HTV | 38 (21) | 6.9 (4.6) | 11 (6.9) | 5.6 (1.8) | 6.2 (2.5) | <0.001 | 0.68 |
| **Neutrophils/ml x10⁴** | LTV | 1.4 (0.72) | 0.91 (0.47) | 1.5 (0.90) | 2.6 (1.8) | 2.1 (0.89) | 0.09 | |
| | HTV | 1.3 (1.2) | 1.5 (1.5) | 11 (11) | 20 (14) | 22 (12) | 0.001 | <0.001 |
| **Lymphocytes/ml x10⁴** | LTV | 3.0 (2.5) | 1.5 (1.6) | 1.9 (1.7) | 1.5 (0.87) | 1.7 (1.5) | 0.54 | |
| | HTV | 3.0 (2.5) | 0.64 (0.75) | 1.3 (1.2) | 1.3 (1.2) | 2.6 (3.2) | 0.09 | 0.25 |
| **Eosinophils/ml x10⁴** | LTV | 0.01 (0.01) | 0.02 (0.05) | 0.02 (0.05) | 0.00 (0.00) | 0.02 (0.04) | 0.80 | |
| | HTV | 0.03 (0.06) | 0.01 (0.02) | 0.11 (0.18) | 0.12 (0.16) | 0.03 (0.04) | 0.18 | 0.30 |

Values displayed as mean (standard deviation). p† indicates result of a one-way ANOVA for each group and variable over time. p# indicates 2-way ANOVA result for treatment group and all time points. LTV – low tidal volume ventilation, HTV – high tidal volume ventilation, Recovery – fraction of instilled bronchoalveolar lavage volume recovered with suction expressed as percent of instilled volume.

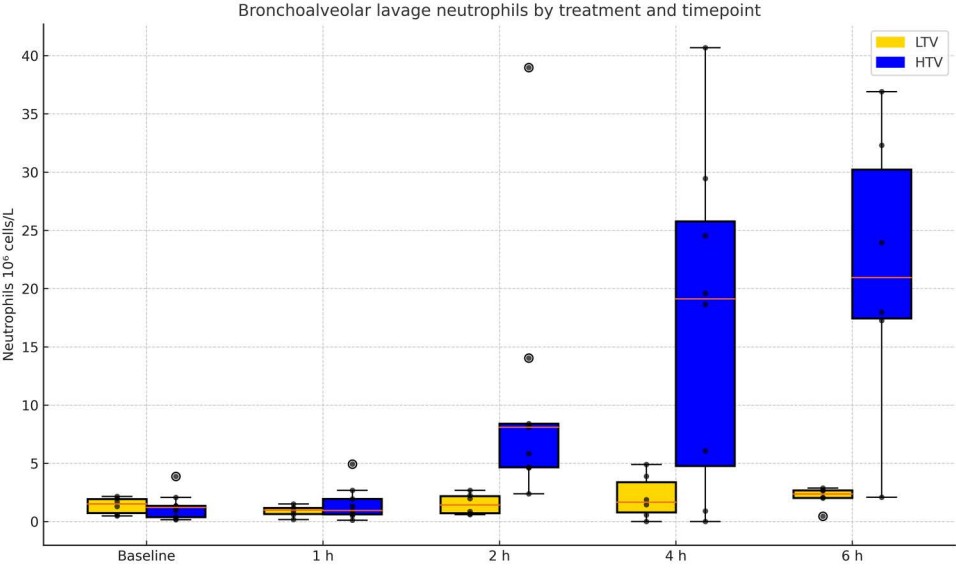

**Fig 2. Neutrophils in bronchoalveolar lavage fluid.** Neutrophil count in bronchoalveolar lavage fluid in high (HTV) and low (LTV) tidal volume group, respectively, over time. The neutrophil count increased over time in the HTV group and was higher compared to the LTV group (numerical results for all cell types and counts in Table 3, and figures for all the other BAL cell results corresponding to the table in S2 Fig).

SYTO 13 and HMGB1 also increased over time in the HTV group (p = 0.001) but not in the LTV group, and the HTV group was higher over time than the LTV group (p = 0.002) (Table 4 and Fig 4). EVs in BALF expressing lactadherin or combinations of lactadherin and CD 61, CD 62 P/E, CD 45 or CD 14, respectively, showed no changes over the duration of the experimental exposure (Table 4, and figures corresponding to the table in S1 Fig). There was no increase in these same EV expression measures over time in plasma in the HTV group (Table 5, and figures corresponding to the table in S3 Fig).

**Table 4. Extracellular vesicles in bronchoalveolar lavage fluid.**

|  |  | Baseline | 1 hour | 2 hours | 4 hours | 6 hours | p† | p# |
|---|---|---|---|---|---|---|---|---|
| **SYTO 13** | LTV | 814 (115) | 796 (114) | 783 (81) | 728 (85) | 827 (126) | 0.63 | |
|  | HTV | 696 (102) | 879 (84) | 932 (205) | 940 (138) | 958 (68) | 0.004 | 0.02 |
| **SYTO 13 + HMGB1** | LTV | 479 (49) | 434 (34) | 414 (46) | 419 (90) | 416 (24) | 0.30 | |
|  | HTV | 409 (44) | 450 (40) | 478 (32) | 501 (51) | 459 (24) | 0.001 | 0.002 |
| **Lactadherin** | LTV | 1453 (322) | 1966 (1021) | 2300 (676) | 1761 (258) | 1616 (358) | 0.23 | |
|  | HTV | 1606 (234) | 2136 (929) | 1645 (262) | 1495 (186) | 1749 (311) | 0.13 | 0.23 |
| **Lactadherin + CD61** | LTV | 174 (25) | 173 (20) | 181 (22) | 167 (21) | 178 (15) | 0.84 | |
|  | HTV | 153 (39) | 153 (30) | 157 (43) | 162 (33) | 168 (21) | 0.93 | 0.92 |
| **Lactadherin + CD62 P/E** | LTV | 63 (18) | 74 (16) | 69 (8) | 67 (20) | 65 (3) | 0.80 | |
|  | HTV | 59 (16) | 61 (19) | 65 (15) | 55 (12) | 65 (13) | 0.72 | 0.85 |
| **Lactadherin + CD45** | LTV | 187 (55) | 215 (73) | 200 (33) | 185 (34) | 159 (70) | 0.61 | |
|  | HTV | 177 (69) | 180 (40) | 184 (44) | 158 (31) | 170 (40) | 0.86 | 0.87 |
| **Lactadherin + CD14** | LTV | 29 (13) | 31 (4) | 32 (8) | 28 (5) | 37 (5) | 0.39 | |
|  | HTV | 23 (9) | 28 (5) | 29 (5) | 26 (6) | 31 (9) | 0.35 | 0.94 |

Extracellular vesicles in bronchoalveolar lavage fluid at different time points displayed as mean (standard deviation). LTV – low tidal volume ventilation. HTV – high tidal volume ventilation. p† indicates result of a one-way ANOVA for each group and variable over time. p# indicates 2-way ANOVA result for treatment group and all time points.

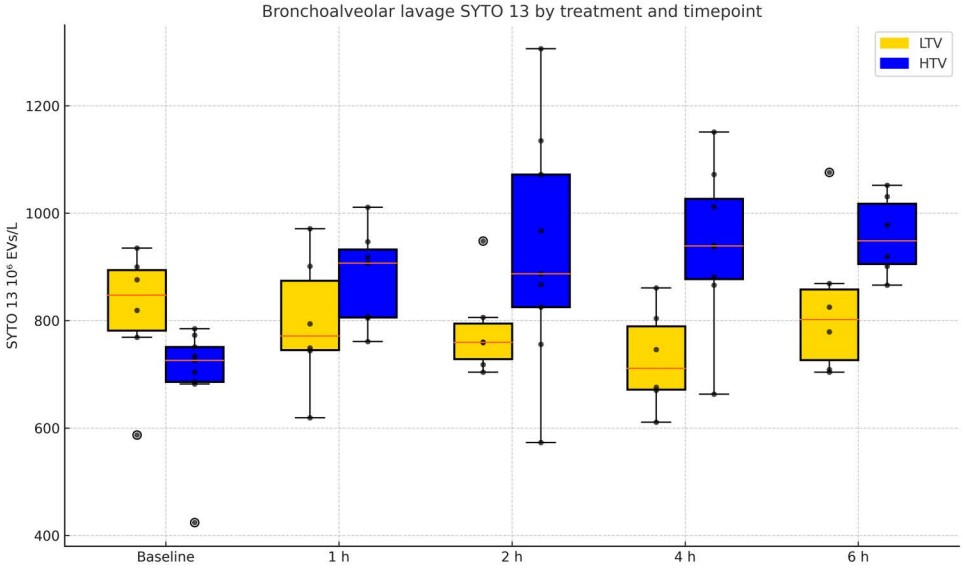

**Fig 3. Extracellular vesicles (EVs) expressing SYTO 13 in bronchoalveolar lavage fluid.** SYTO 13 positive extracellular vesicles (EVs) in high (HTV) and low (LTV) tidal volume group, respectively, over time. SYTO 13 positive EVs increased over time in the HTV group and was higher compared to the LTV group (numerical results in Table 4).

## Discussion

The main findings were that experimental hyperinflation condition was associated with higher levels of EVs marked by SYTO 13, as well as the combination of SYTO 13 + HMGB1, in bronchoalveolar lavage fluid, but not in blood for this early injury time interval. In the group not exposed to injurious high tidal volumes, this pattern of increased EV expression was not

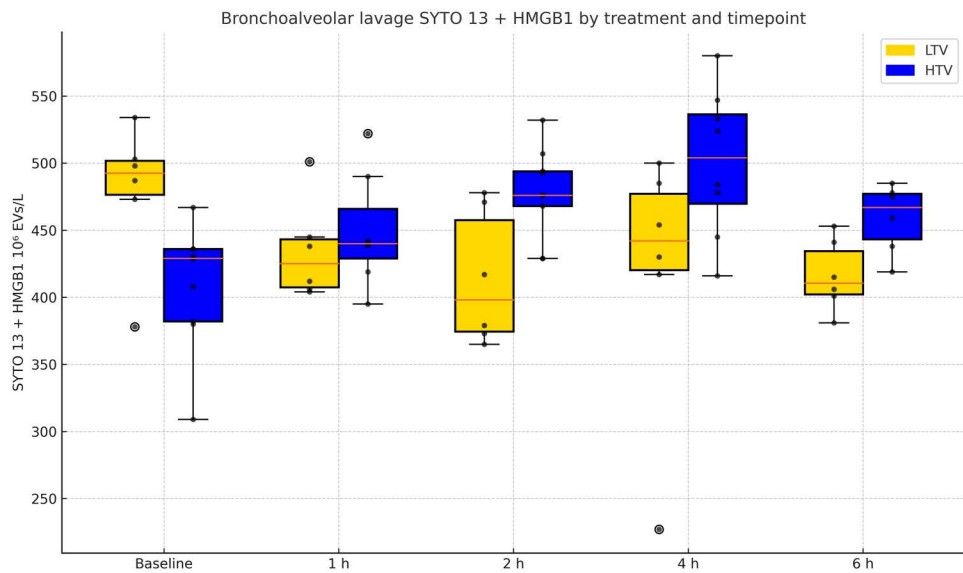

**Fig 4. Extracellular vesicles (EVs) expressing SYTO 13 + HMGB1 in bronchoalveolar lavage fluid.** Extracellular vesicles positive for SYTO 13 and HMGB1 in bronchoalveolar lavage fluid in high (HTV) and low (LTV) tidal volume group, respectively, over time. SYTO 13 and HMGB1 positive EVs increased over time in the HTV group and was higher compared to the LTV group (numerical results in Table 4).

**Table 5. Extracellular vesicles in plasma.**

| | | Baseline | 1 hour | 2 hours | 4 hours | 6 hours | p† | p# |
|---|---|---|---|---|---|---|---|---|
| SYTO 13 | LTV | 398 (53) | 427 (46) | 404 (44) | 418 (67) | 425 (87) | 0.92 | |
| | HTV | 382 (65) | 435 (96) | 486 (107) | 424 (76) | 404 (51) | 0.17 | 0.46 |
| SYTO 13 + HMGB1 | LTV | 314 (13) | 347 (13) | 339 (23) | 363 (39) | 373 (33) | 0.02 | |
| | HTV | 364 (50) | 340 (49) | 372 (52) | 344 (40) | 329 (22) | 0.40 | 0.03 |
| Lactadherin | LTV | 1922 (175) | 2006 (292) | 2211 (467) | 2100 (375) | 2056 (203) | 0.69 | |
| | HTV | 1834 (276) | 1992 (284) | 1792 (621) | 2338 (696) | 1989 (132) | 0.22 | 0.43 |
| Lactadherin + CD61 | LTV | 140 (17) | 135 (71) | 139 (85) | 183 (40) | 193 (58) | 0.71 | |
| | HTV | 124 (74) | 154 (93) | 163 (127) | 144 (95) | 129 (37) | 0.92 | 0.74 |
| Lactadherin + CD62 P/E | LTV | 71 (18) | 77 (22) | 90 (25) | 70 (34) | 82 (14) | 0.66 | |
| | HTV | 65 (12) | 78 (22) | 86 (24) | 88 (22) | 76 (13) | 0.15 | 0.60 |
| Lactadherin + CD45 | LTV | 100 (27) | 89 (29) | 93 (36) | 102 (49) | 102 (26) | 0.96 | |
| | HTV | 110 (37) | 104 (30 | 98 (35) | 98 (35) | 112 (40) | 0.91 | 0.97 |
| Lactadherin + CD14 | LTV | 8 (2) | 11 (3) | 10 (3) | 10 (5) | 11 (3) | 0.68 | |
| | HTV | 10 (3) | 11 (3) | 11 (8) | 9 (4) | 11 (3) | 0.89 | 0.89 |

Extracellular vesicles in plasma at different time points displayed as mean (standard deviation). LTV – low tidal volume ventilation. HTV – high tidal volume ventilation. p† indicates result of a one-way ANOVA for each group and variable over time. p# indicates 2-way ANOVA result for treatment group and all time points.

observed. No increases in EVs in BALF identified by other markers were observed for either treatment group. This supports the idea that it may be possible to find markers of acute injury in EV expression, in this case lung volutrauma or stretch injury.

Extracellular vesicles (EVs) have been previously identified as significant contributors to the pathophysiology of acute lung injury [13], with studies highlighting their role in mediating inflammatory responses and cellular communication in lung tissues [17]. In the injury group

of this study, a marked increase in the expression of extracellular vesicles was observed in tandem with the progression of lung injury. This elevation in EV levels could be linked to the presence of inflammatory cells infiltrating the alveolar spaces, suggesting that these vesicles may play a crucial role in the inflammatory cascade that characterizes acute lung injury. In the injury group, EV expression increased as the injury progressed as demonstrated by inflammatory cells in the samples from alveoli and distal airways.

Moreover, the role of extracellular vesicles (EVs) in mediating inflammatory responses extends beyond just acute lung injury; they are also implicated in chronic respiratory conditions such as chronic obstructive pulmonary disease (COPD) and asthma. Studies have shown that EVs can transfer pro-inflammatory cytokines and microRNAs between immune cells and structural cells within the lungs, thereby exacerbating inflammation and tissue damage over time [18].

A pattern change between groups in EV expression could not be detected in blood sampling, though this sampling was only for the first few hours of injury. This sampling period was chosen because the two-hit volutrauma injury porcine model is harsh and does not allow a stable sampling platform further out in time. The aim using this model was to see if there was a reliable blood plasma marker of this rapidly developing acute lung injury, and this in the first hours. In future studies, extending the sampling duration may provide insights into the temporal dynamics of biomarker changes and their correlation with injury severity.

The BALF EV expressions were not detected in plasma samples. One potential explanation for this absence could be the timing of the plasma sampling; it is conceivable that the samples were obtained too early in the process, assuming that alveolar-derived EVs would eventually migrate into the bloodstream over time. This suggests a temporal dynamic in the release and circulation of these vesicles that may not have been captured in the initial sampling. This notion of delayed EV detection in plasma raises intriguing questions about the mechanisms governing their release and transport post-injury. For instance, recent studies have shown that a heterogeneous population of EVs is mobilized to the alveoli after severe injuries, suggesting that local microenvironments significantly influence EV dynamics [19]. If this is indeed the case, it may be essential to consider not only the timing of blood sampling but also the specific injury context when interpreting EV profiles. Understanding how these vesicles migrate from the site of injury into systemic circulation could illuminate potential therapeutic targets for mitigating acute lung injury.

Alternatively, it is possible that a much higher degree of EV expression might be needed before the same EVs come into the blood stream. For clinical purposes, plasma sampling to follow the course of lung injury progression would be valuable as BALF sampling is highly invasive, though there are findings that suggest that substances from the lower airways, which are aerosolized, can be sampled noninvasively [20,21]. The findings from this study of the first hours of acute volutrauma-related lung injury do not support plasma diagnostic sampling in this context. Also, changes in plasma levels of EVs may be disguised by EV's from cells native to the blood stream, as well as by other particles such as lipoproteins, platelets and other particles [22].

Since SYTO 13 stains both DNA and RNA, these findings show that EVs in BALF containing nucleic acid materiel increase but without distinction of which forms of nucleic acids. Further, increase in EVs in BALF with nucleic acid and HMGB1 during progressive injury was observed but was less distinct. The role of nucleic acids in EVs remains unclear, but HMGB1 is widely recognized as a damage-associated molecular pattern marker (DAMP).

A two-hit injury model was used to produce a rapidly developing volutrauma lung injury, confirmed by leukocyte recruitment (mostly neutrophils) to the alveolar space. Even the surfactant depletion comparison group sustains an injury, though milder and without obvious progression in this time frame. The calibrated volutrauma injury was severe and provided a good opportunity

to test if injury signals in the lungs are quickly detectable in blood. While hyperinflation was the primary experimental intervention, the model also include high airway pressures that may cause barotrauma, along with hyperoxia and the absence of PEEP that may cause atelectasis and possible atelectotrauma [23]. In this way, the acute injury might have been multimodal.

One limitation of the study design is that porcine acute lung injury probably does not translate completely to human acute lung injury. Additionally, this very potent volutrauma injury is probably much more severe than volutrauma that generally occurs in the ICU context. The very high inspired oxygen level in the HTV group was designed to prevent hypoxemia, though where lower inspired oxygen levels in the LTV group potentially could introduce bias in the results related to oxygen toxicity. The EV and EV content markers used were limited, but probably adequate to answer the question about appearance of plasma EVs related to acute lung injury in this time interval. Concerning cell origin of EVs, it is likely that some, or many, came from neutrophils (CD45 consistent with neutrophils but not neutrophil-specific), but EV expression from other cell types is possible. Neutrophil-specific markers such as CD66b or CD15 or MPO were not included.

In summary, EVs containing nucleic acids and HMGB1 were detected in bronchoalveolar lavage fluid early in severe hyperinflation-related acute lung injury, but not in simple surfactant depletion without hyperinflation. EV's with these markers were not observed to increase within the same time frame in plasma.

## Conclusion

We conclude that these findings do not support the ability to detect early severe acute lung injury through EVs sampled in plasma, at least in this first hours' timeframe. EVs containing nucleic acids may increase in the alveolar space during the early progression of volutrauma.

## Supporting information

**S1 File. Raw data.** All raw data are included in this Excel file.
(XLSX)

**S1 Fig. EVs in BALF.** Panels of EVs positive for lactadherin and lactadherin in combination with CD14, CD45, CD61 and CD62P/E, respectively.
(PDF)

**S2 Fig. BALF cells.** Cell counts of total cells, eosinophils, lymphocytes and macrophages as well as recovery rate.
(PDF)

**S3 Fig. EVs in plasma.** Panels of EVs positive for SYTO 13, SYTO 13 combined with HMGB1, lactadherin and lactadherin in combination with CD14, CD45, CD61 and CD62P/E, respectively.
(PDF)

## Author contributions

**Conceptualization:** Niklas Larsson, Jonas Claesson, Stefan Lehtipalo.

**Formal analysis:** Niklas Larsson, Michael Haney.

**Funding acquisition:** Niklas Larsson.

**Investigation:** Niklas Larsson, Jonas Claesson, Stefan Lehtipalo, Fariborz Mobarrez.

**Methodology:** Niklas Larsson, Jonas Claesson, Stefan Lehtipalo, Annelie Behndig, Fariborz Mobarrez.

**Supervision:** Stefan Lehtipalo, Michael Haney.

**Writing – original draft:** Niklas Larsson, Fariborz Mobarrez, Michael Haney.

**Writing – review & editing:** Niklas Larsson, Jonas Claesson, Stefan Lehtipalo, Annelie Behndig, Fariborz Mobarrez, Michael Haney.

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
