## [Decision Letter · Decision Letter 0]

15 Dec 2024

PONE-D-24-49430Extracellular vesicle release in an experimental ventilator-induced lung injury porcine modelPLOS ONE

Dear Dr. Larsson,

Thank you for submitting your manuscript to PLOS ONE. After careful consideration, we feel that it has merit but does not fully meet PLOS ONE’s publication criteria as it currently stands. Therefore, we invite you to submit a revised version of the manuscript that addresses the points raised during the review process.

Thank you for submitting your manuscript to [Journal name]. After careful consideration and review of the reviewer comments, I am writing to inform you that a revision of your manuscript is required before we can consider it further for publication.

The reviewers have raised several important points that need to be addressed. Specifically, we request that you either conduct additional experiments to support your findings or provide a comprehensive rebuttal to the reviewers' concerns. These revisions are essential to strengthen the scientific merit of your work.

To proceed with your submission, please:

Address each reviewer comment systematicallyEither perform the suggested additional experiments where necessary, orProvide clear, well-supported arguments explaining why such experiments may not be needed

When submitting your revised manuscript, please include a point-by-point response to the reviewers' comments, clearly indicating how you have addressed each concern.

We look forward to receiving your revised manuscript.

Kind regards,

Rui Tada, Ph.D.

Academic Editor

PLOS ONE

 Journal Requirements: When submitting your revision, we need you to address these additional requirements. 1. Please ensure that your manuscript meets PLOS ONE's style requirements, including those for file naming. The PLOS ONE style templates can be found at https://journals.plos.org/plosone/s/file?id=wjVg/PLOSOne_formatting_sample_main_body.pdf and https://journals.plos.org/plosone/s/file?id=ba62/PLOSOne_formatting_sample_title_authors_affiliations.pdf 2. Please include a caption for figure 1 and 5.  3. We note that the grant information you provided in the ‘Funding Information’ and ‘Financial Disclosure’ sections do not match.  When you resubmit, please ensure that you provide the correct grant numbers for the awards you received for your study in the ‘Funding Information’ section. 4. Thank you for stating the following financial disclosure:  [This work was supported by the ALF-LUA cooperative Region Västerbotten-Umeå University intramural research funding system.].  Please state what role the funders took in the study.  If the funders had no role, please state: ""The funders had no role in study design, data collection and analysis, decision to publish, or preparation of the manuscript."" If this statement is not correct you must amend it as needed. Please include this amended Role of Funder statement in your cover letter; we will change the online submission form on your behalf. 5. Thank you for stating the following in the Competing Interests section: [Niklas Larsson has received speaker fees for lecturing in clinical symposia sponsored by Dräger Medical, unrelated to this specific topic.].  Please confirm that this does not alter your adherence to all PLOS ONE policies on sharing data and materials, by including the following statement: ""This does not alter our adherence to  PLOS ONE policies on sharing data and materials.” (as detailed online in our guide for authors http://journals.plos.org/plosone/s/competing-interests).  If there are restrictions on sharing of data and/or materials, please state these. Please note that we cannot proceed with consideration of your article until this information has been declared.  Please include your updated Competing Interests statement in your cover letter; we will change the online submission form on your behalf.

Reviewers' comments:

Reviewer's Responses to Questions

**Comments to the Author**

1. Is the manuscript technically sound, and do the data support the conclusions?

Reviewer #1: Yes

Reviewer #2: No

2. Has the statistical analysis been performed appropriately and rigorously? 

Reviewer #1: Yes

Reviewer #2: Yes

3. Have the authors made all data underlying the findings in their manuscript fully available?

Reviewer #1: Yes

Reviewer #2: Yes

4. Is the manuscript presented in an intelligible fashion and written in standard English?

Reviewer #1: Yes

Reviewer #2: Yes

5. Review Comments to the Author

Reviewer #1: The authors have conducted an in-vivo study in pigs to evaluate how high tidal volume ventilation after surfactant depletion alters the generation of extracellular vessicles (EVs) in the bronchiolar lavage fluid and in the plasma. They also monitor a marker of lung damage and demonstrate that nucleic acid containing EVs increase in bronchoalveolar lavage fluid but not in the plasma. This is an important finding as it indicates that airspace markers might be more specific for lung injury during mechanical ventilation and that plasma biomarkers may not be indicated for assessing the degreee of lung injury during ventilation. This is an important study that I recommend for publication but I do have several comments that should be addressed prior to publication.

1. My main comment is that the ventilation protocol used, high tidal volume following lung lavage/surfactant inactivation may involve other forms of lung injury besides stretching injury. First, surfactant depletion will cause airway/alveolar atelectasis and even ventilation at the PEEP levels may result in cyclic airway closure and reopening. Recently, this form of injury (atelectrauma) has been implicated as a more damaging force than stretching, see Gabela-Zuniga, Lab on Chip, 2024. Second, the data in Table 2 indicate that there is higher inspiratory and airway pressures in the HTV groups and therefore transmural induced inflammation (barotrauma) may also be occuring. The authors should at minimum note the importance of these other forms of ventilation induced lung injury and note that they may be occuring to some degree in thier animal model.

2. I suggest referencing the novel work by Shaver et al in which protein biomarkers of ARDS were obtained from heat moisture exchange (HME) filter fluid as a way of assessing airway markers. Extension of this technique to obtain nucleic acids from HME fluid might be an important new way to non-invasively assess lung injury markers from the airspace and thus more specifically assess the degree of lung injury in mechanically ventilated patients See Bastarache, Am J Physiol Lung Cell Mol Physiol. 2021 and Rizzo, JCI Insight. 2022.

3. Since the authors use 100% O2 in the HTV group, there could also be hyperoxic lung injury. This should be noted.

4. It should be noted that BAL could contain EVs from both airway as well as alveolar compartments

5. Given that neutrophil counts are going up in the presented model of VILI, it is possible that the increased EV counts seen in BAL are due to these cells. Is there any evidence for which cells these EVs come from? Surface markers?

6. I think table 5 would be better served as a figure (like figure 3-4)

7. Finally, in addition to being a biomarker of disease, it might be worth mentioning that EVs have recently been used as a therapeutic drug delivery vehicle to minize VILI see Salazar‐Puerta, Advanced Materials, 2023.

Reviewer #2: Thank you for asking me to review this paper. Larssen et al use a porcine model of saline lung lavage and high stretch to examine release of extracellular vesicles in relation of VILI. They show increases in bronchoalveolar lavage neutrophils, and EVs containing SYTO-13 and HMGB-1 combined. I have a few comments:

1. I do not think the data add to the knowledge base. It is known that EVs and neutrophils are increased in ARDS/VILI.

2. Further investigation as to where the EVs derived from would be useful, given the use of flow cytometry.

3. Neutrophil EVs are associated with SYTO-13 especially if derived during NETosis induced DNA release.

4. HMGB-1 is also associated with NET release

5. Increased neutrophil numbers in the BAL would also mean higher EVs being detected. Especially given other EVs e.g. platelet derived ones did not increase.

6. What are the downstream effects of the EVs that have been detected?

6. PLOS authors have the option to publish the peer review history of their article (what does this mean? ). If published, this will include your full peer review and any attached files.

**Do you want your identity to be public for this peer review?** For information about this choice, including consent withdrawal, please see our Privacy Policy .

Reviewer #1: No

Reviewer #2: No

---

## [Author Response · Author response to Decision Letter 1]

21 Dec 2024

Thank you for your review and constructive comments!

Journal Requirements

1 The manuscript has been revised to confirm to the style templates and we have tried to catch all the details.

2. Caption for figure 1 in included in rows 137-140. There is no figure 5 and the erroneous reference to fig 5 has been removed from the manuscript.

3 We have updated the financial disclosure statement. The organizations Västerbottens Läns Landsting and Umeå University have a cooperative grant arrangement and are linked in this context. We apologize for the previous unclarity.

4. The funders had no role in study design, data collection and analysis, decision to publish, or preparation of the manuscript

5. We have entered the requested statement as a comment in the comment section above.

Response to reviewer #1

1We agree in full and have added this to the discussion in rows 346-349.

2We agree that this is important. We have added a short reference to noninvasive sampling to the discussion in rows 328-329.

3 This is correct and we have added this to the discussion together with a comment on other possible injury mechanisms (row 346-349).

4. We agree and we have added this to the methods part (row 145-146) and to the discussion (row 294)

5.In our study, the increased neutrophil counts in BAL support the hypothesis that the elevated EV levels could partially originate from these cells as you mention. While our study used markers like SYTO 13 and HMGB1 to indicate nucleic acid content and cellular stress, these are not specific for neutrophils. CD45, a general leukocyte marker included in the study, may also encompass EVs derived from neutrophils, though it lacks the specificity to confirm this conclusively. Neutrophil-specific markers such as CD66b or CD15 or MPO were not included. Future studies incorporating such markers would help definitively confirm the contribution of neutrophils to the EV pool observed in BAL. A short reference to this was added to the limitations paragraph in discussion (rows 358-361).

6. We believe the most relevant data is highlighted by limiting figures in the manuscript to the variables with significant changes in the intervention group. We have, however, included figures of all EVs and BALF cells in supplements to make overview easier. However, if the Editor wants us to, we will be happy to replace tables 4 and 5 with multi-panel figures like figure 3 and 4. Otherwise, we have updated supplements with S1-3 fig.

7This is indeed a very interesting development. However, we feel that the scope of this paper is limited to biomarker and diagnostic aspects. Therapeutic aspects regarding VILI are possibly beyond the scope of the current paper.

Reviewer #2

1. The main study question was if EVs, which appear in the lungs, in connection with acute lung injury, will be detectable in the first hours in plasma. We feel that this is a novel question and finding, although we cannot rule out plasma as a sampling source for early acute lung injury, our findings do not suggest that there is a large and rapid appearance of EVs from a specific organ injury in the early phase.

2. We agree that further investigations of the EVs we studied would be very interesting. Unfortunately, we do not have more samples to analyze but EV parental cells in this context are very relevant for analysis in future studies.

3. While NETosis in this model is possible, we have not studied it specifically. SYTO-13 can indeed stain EVs associated with NETosis due to its binding to extracellular DNA. However, it is not specific to NETs or neutrophils, as it also binds to EVs from other cell types. In our previous studies, we have observed that platelets and platelet derived EVs exhibit strong SYTO-13 binding, likely due to their mRNA content. While the current findings could partially reflect neutrophil activity during NETosis, further studies specifically targeting neutrophil EVs and NETosis markers are needed to clarify this.

(Ref: https://pubmed.ncbi.nlm.nih.gov/31103269/ )

4. HMGB-1 is indeed associated with NET release and could reflect neutrophil activity in this context. However, it is not specific to NETs or neutrophils, as HMGB-1 is also released by other cell types, including platelets, as documented in the literature. While HMGB1 is associated with cellular stress or damage, it cannot solely be attributed to NETosis or neutrophils. Future studies using more specific NET and neutrophil markers would be essential to better delineate the role of HMGB-1 in this context.

5. The increased neutrophil numbers in BAL likely contribute to the higher EV levels observed. However, it would be too speculative to definitively attribute the observed EV increase solely to neutrophils without the use of additional specific markers. Platelets are relevant as possible parent cells.

6. In the given context, any statement about downstream effects would be speculative, as this study does not analyze or establish functional effects of the EVs observed here. However, it is well-documented that EVs can have both pro-inflammatory and anti-inflammatory effects, depending on their cargo and other characteristics, which includes a variety of proteins, nucleic acids, and other bioactive molecules. These effects could influence inflammation, immune modulation, and tissue remodeling. Determining the specific downstream effects of the EVs observed in this study would require additional experimental approaches, such as functional assays or proteomic and transcriptomic analyses of EV cargo and pathway analysis.

---

## [Decision Letter · Decision Letter 1]

14 Feb 2025

Extracellular vesicle release in an experimental ventilator-induced lung injury porcine model

PONE-D-24-49430R1

Dear Dr. Larsson,

We’re pleased to inform you that your manuscript has been judged scientifically suitable for publication and will be formally accepted for publication once it meets all outstanding technical requirements.

Kind regards,

Rui Tada, Ph.D.

Academic Editor

PLOS ONE

Additional Editor Comments (optional):

Reviewers' comments:

Reviewer's Responses to Questions

**Comments to the Author**

1. If the authors have adequately addressed your comments raised in a previous round of review and you feel that this manuscript is now acceptable for publication, you may indicate that here to bypass the “Comments to the Author” section, enter your conflict of interest statement in the “Confidential to Editor” section, and submit your "Accept" recommendation.

Reviewer #1: All comments have been addressed

2. Is the manuscript technically sound, and do the data support the conclusions?

Reviewer #1: Yes

3. Has the statistical analysis been performed appropriately and rigorously? 

Reviewer #1: Yes

4. Have the authors made all data underlying the findings in their manuscript fully available?

Reviewer #1: Yes

5. Is the manuscript presented in an intelligible fashion and written in standard English?

Reviewer #1: Yes

6. Review Comments to the Author

Reviewer #1: (No Response)

7. PLOS authors have the option to publish the peer review history of their article (what does this mean? ). If published, this will include your full peer review and any attached files.

**Do you want your identity to be public for this peer review?** For information about this choice, including consent withdrawal, please see our Privacy Policy .

Reviewer #1: No

---

## [Editor Report · Acceptance letter]

PONE-D-24-49430R1

PLOS ONE

Dear Dr. Larsson,

I'm pleased to inform you that your manuscript has been deemed suitable for publication in PLOS ONE. Congratulations! Your manuscript is now being handed over to our production team.

Kind regards,

on behalf of

Dr. Rui Tada

Academic Editor

PLOS ONE